# Preferences for different diagnostic modalities to follow up abnormal colorectal cancer screening results: a hypothetical vignette study

Aradhna Kaushal [1], Sandro Tiziano Stoffel,[1,2] Robert Kerrison [1], Christian von Wagner [1]

¹Research Department of Behavioural Science and Health, University College London, London, UK
²European Center of Pharmaceutical Medicine, University of Basel, Basel, Switzerland

**Correspondence to**
Dr Christian von Wagner;
c.wagner@ucl.ac.uk

## ABSTRACT

**Objectives** In England, a significant proportion of people who take part in the national bowel cancer screening programme (BCSP) and have a positive faecal occult blood test (FOBt) result, do not attend follow-up colonoscopy (CC). The aim of this study was to investigate differences in intended participation in a follow-up investigation by diagnostic modality offered including CC, CT colonography (CTC) or capsule endoscopy (CE).

**Setting** We performed a randomised online experiment with individuals who had previously completed an FOBt as part of the English BCSP.

**Methods** Participants (n=953) were randomly allocated to receive one of three online vignettes asking participants to imagine they had received an abnormal FOBt result, and that they had been invited for a follow-up test. The follow-up test offered was either: CC (n=346), CTC (n=302) or CE (n=305). Participants were then asked how likely they were to have their allocated test or if they refused, either of the other tests. Respondents were also asked to cite possible emotional and practical barriers to follow up testing. Multivariable logistic regression models were used to investigate intentions.

**Results** Intention to have the test was higher in the CTC group (96.7%) compared with the CC group (91.8%; OR 2.64; 95% CI 1.22 to 5.73). CTC was considered less 'off-putting' (OR 0.66, 95% CI 0.47 to 0.94) and less uncomfortable compared with CC (OR 0.51, 95% CI 0.34 to 0.77). For those who did not intend to have the test they were offered, CE (39.7%) or no investigation (34.5%) was preferable to CC (8.6%) or CTC (17.2%).

**Conclusions** Alternative tests have the potential to increase attendance at diagnostic follow-up appointments.

## INTRODUCTION

Colorectal cancer (CRC) is the fourth most common cancer in the UK, accounting for 12% of all cancer diagnosis, and is the second-leading cause of cancer death.[1] Screening can reduce mortality and improve survival by detecting CRC at an earlier stage, when treatment is more likely to be successful.[2] In England, the national bowel cancer screening programme sends a screening test to all men

and women aged 60–74 to be completed at home. The test is offered once every 2 years, and can detect small amounts of blood in the stool, which may be indicative of CRC. This test until recently was the 'faecal occult blood test' (FOBt) and was replaced with the more sensitive 'feacal immunochemical test' in June 2019. Around 2% of people who complete an FOBt have a positive test result and are invited for further investigation, usually a colonoscopy (CC). Despite being at increased risk, approximately 14% of those with an abnormal test result fail to complete their diagnostic follow-up.[3–6] There are a number of reasons why a CC in a small proportion of these patients would not be appropriate, such as clinical decisions based on frailty, or having had a recent CC outside the BSCP.[7 8]

► Use of a vignette design allowed us to investigate attitudes towards hypothetical scenarios which would be more difficult to carry out in a real-world setting.
► The use of comprehension checks after presenting vignettes ensured that participants engaged with materials and had an understanding of the tests they were presented.
► The study design presented a hypothetical scenario and so it is not possible to say if intentions would align with behaviours in real life.
► We only provided brief descriptions of the test which may have biased intentions.
► The online panel used in the survey had a very small number of Black, Asian and minority ethnic participants, and therefore, may not be representative of the UK population.

Previous research point towards a range of psychological and practical factors which may influence decisions to have CRC screening.[8–10] Plumb *et al* reviewed the medical records of

patients of 170 patients to identify patient factors associated with non-attendance at CC after a positive FOBt and found the most cited reason was 'unwillingness to complete the test'.[8] For some patients, this was due to anxiety about pain, and risks associated with screening but in the majority of cases, the reason was unspecified. This is supported by a recent systematic review on incomplete diagnostic follow-up after an abnormal CRC screening result which identified perception of pain as the most commonly cited reason in addition to 'embarrassment' and 'being too busy'.[11] Consequently, for a vast majority of this group of patients, there is a strong need to develop patient or system level interventions to remove modifiable barriers to investigating the cause of their FOBt result.[12]

Some of these barriers may be directly or indirectly related to the nature of CC. CC involves inserting a long flexible tube with a tiny camera on the end into the anus in order to examine the bowel. If polyps (small, potentially cancerous growths) are found, they are removed during the procedure. The test requires the patient to restrict their diet, take a strong laxative and arrange for someone to take them home afterwards due to drowsiness caused by a sedative. After the procedure, most people will need to rest for the remainder of the day.[13]

Considering the practical and emotional barriers which may inhibit someone from accepting the offer of further investigation via CC, it has been suggested that offering alternative tests, such as CT colonography (CTC) or capsule endoscopy (CE), may be more acceptable to patients.[14–16] These tests differ in terms of preparation, time in hospital and level of perceived invasiveness. Both of these tests, as with CC, require the patient to restrict their diet and take a laxative. In CTC, X-rays are used to take images while a small tube is used to inflate the bowel. After CTC, most people are able to resume their daily activities.[17] In CE, the patient ingests a capsule containing a small camera which takes photos inside the bowel and transmits them wirelessly to a receiver worn by the patient.[18] Patients are able to carry on with their normal activity and the capsule is passed through the body after 8 hours. For both of these tests, a CC is usually recommended to remove any polyps if they are found.

CTC and CE could be offered individuals who decline to have a CC after a positive FOBt, in the expectation that providing individuals with alternative choices increases their feeling of autonomy and intrinsic motivation.[19–21] While offering choice seems promising, there exists no evidence that offering more than one screening test influences adherence and patient satisfaction.[22] Although CTC and CE are not currently endorsed as a screening methods, it is possible that offering alternative tests for those who do stop engaging with the screening programme may increase intentions to have further investigations.

The present study aimed to evaluate whether offering different screening tests, such as CTC or CE, increases intentions to attend for further investigation, when compared with offering a CC.

## METHODS
### Design
We performed a randomised online experiment with a survey company called: 'Dynata' (formerly Survey Sampling International and Research Now). Potentially eligible men and women were invited to participate in the online experiment by Dynata, who invited them from their panel if they were: (1) aged 60–74 years and (2) lived in England. Potentially eligible men and women were informed that they would be rewarded with 'Dynata points', which they could exchange for money, or a donation to a charity, if they complete the survey.

Individuals who agreed to participate in the survey were asked about their screening history, which was used to filter ineligible adults. Only individuals who had completed an FOBt previously, and received a normal result, were eligible to participate in the survey. We excluded participants who had previously had a CC, a previous diagnosis of CRC or had parts of their bowel removed. Once eligibility to participate had been established using these criteria, participants were then randomly allocated to one of three experimental conditions in which they were asked to read a hypothetical vignette (see online supplementary appendix 1).

Depending on the condition individuals were allocated to, the vignette asked them to imagine that their next FOBt result was abnormal, and they were being invited for a follow-up test consisting of either CC, a CTC or a CE. Each vignette included a some images of the test, a short description about what happens before, during and after the test, and any risks associated with the offered test. After reading the information, participants were required to successfully answer a 'comprehension check question' about the test (what happens during the test, how long it takes, and what happens afterwards), to ensure they had understood the information. If answered incorrectly, the information about the test was presented again and the participant is provided with another opportunity to answer the question. Once the participant answered the comprehension check question correctly, they were asked about their intention to have the follow-up test. Details of the comprehension check questions can be found in the online supplementary appendix 1.

### Measures
#### Intention to have follow-up test
On answering the comprehension check question correctly, participants were asked to indicate their intention to book an appointment ('Considering all the information presented above, would you take up the offer of this test?'), using a four-point Likert scale, with response options: 'definitely not', 'probably not', 'probably, yes' and 'definitely, yes'.[23 24] These responses were dichotomised ('probably, yes' or 'definitely, yes' vs 'probably not'

or 'definitely not') due to the relative lack of 'probably not' and 'definitely not' responses (see online supplementary appendix 2).

### Barriers for follow-up test

After indicating their intentions, participants were then asked to indicate how they felt about potential barriers for not wanting the tests ('Please read each statement and select how strongly you agree or disagree with it?'), using a four-point Likert scale, with response options: 'strongly agree', 'slightly agree', 'slightly disagree' and 'strongly disagree'. The barriers featured six emotional and four practical items, and were derived from previous surveys.[8 25] The six emotional barrier items were: 'The preparation for the test (restricted diet and strong laxative) puts me off', 'The test looks like it would be uncomfortable', 'I would be embarrassed about taking the test', 'I would worry about the risks associated with the test', 'I would be afraid of getting an abnormal result' and 'Doing the test would make me worry more about bowel cancer'. The four practical barrier items were: 'I would not have time to do the test', 'I have other problems to worry about', 'It would be difficult to arrange transport to the hospital' and 'I have other health problems that are more important'. These responses were dichotomised ('strongly agree' or 'slightly agree' vs 'slightly disagree' or 'strongly disagree') due to the relative lack of 'strongly agree' responses (see online supplementary appendix 2).

### Preference of disinclined study participants

Study participants who indicated that they would not (probably or definitely) have the offered follow-up test were presented with information about the other two tests and asked which of the three tests they would prefer ('Which of the two tests would you prefer to have?'), with response options 'CC', 'capsule endoscopy', 'CTC' and 'none of them'.

### Numeracy skills

Numeracy skills were assessed by the question: 'Which of the following numbers represents the biggest risk of getting a disease?', with answer options '1/10', '1/100', '1/1000' and 'I don't know'. This measure was adapted from Lipkus et al,[26] who previously validated the question as a measure for numeracy skills.

### Sociodemographic variables

Details of the participant's age, gender, ethnicity, employment status, education, car and home ownership were collected at the end of the survey. Ethnicity data were recoded as 'White' (White British or Other White Background) and Black, Asian and minority ethnic (BAME) due to the small numbers in each group. The three variables on education, car and home ownership were used to calculate a proxy measure for socioeconomic deprivation. One point was given to a participant if their household did not own a car or van, if they had no formal qualifications and if they did not own their own home.[27–29] Scores,

therefore, ranged from 0 to 3, with high scores indicating higher levels of deprivation.

### Sample size calculation

Sample size of this study was calculated prior to data collection based on the results of a soft launch. We calculated that we needed approximately 300 participants per condition to detect differences of at least 5 percentage points in proportion of intenders effect size between any of the three conditions, with a power of 80% and an alpha value of 0.05.[30]

### Statistical analysis

Our primary outcome was intention to have the offered follow-up test after exposure to the allocated vignette. Our secondary outcomes were the responses to the perceived emotional and practical barrier questions. Descriptive statistics were used to report the sociodemographic characteristics of the study population. Univariable and multivariable logistic regression models were used to investigate the effect of offering alternative follow-up tests on participants' intentions to have further investigations, as well as their perceived emotional and practical barriers of the test offered. Covariates that were included in the regression models were age, gender, ethnicity, deprivation score, employment status and numeracy. All statistical analyses were conducted using Stata/SE (V.15.1).[31] The survey, data and Stata codes for the experiment are available via Open Science Framework:https://osf.io/fx69t/.

### Patient and public involvement

The design of this study followed a direct question posed by representatives of Public Health England and was informed by previous research on this topic and close collaboration with experts involved in administering relevant tests (see Acknowledgments).

## RESULTS
### Participants

In total, 1926 adults responded to the online invitation. Of these, 481 (25.0%) were excluded due to their age or medical history (eg, previous bowel cancer diagnosis or removal of part of the bowel). A further 372 (19.3%) were excluded as they had never been invited for, or completed, an FOBt, and an additional 46 (2.4%) were excluded as they had previously received an abnormal FOBt result. Of the remaining 1027 adults who were eligible for inclusion, 953 (92.8%) completed the survey (see figure 1 for an overview of participants through the study).

Table 1 shows the participant characteristics by study condition. Around half of participants were male (54.5%). Most of the participants were of a White ethnic background (98.6%), not in paid employment (75.1%), had a formal education (62.6%), owned car (87.1%) and owned a house (87.4%).

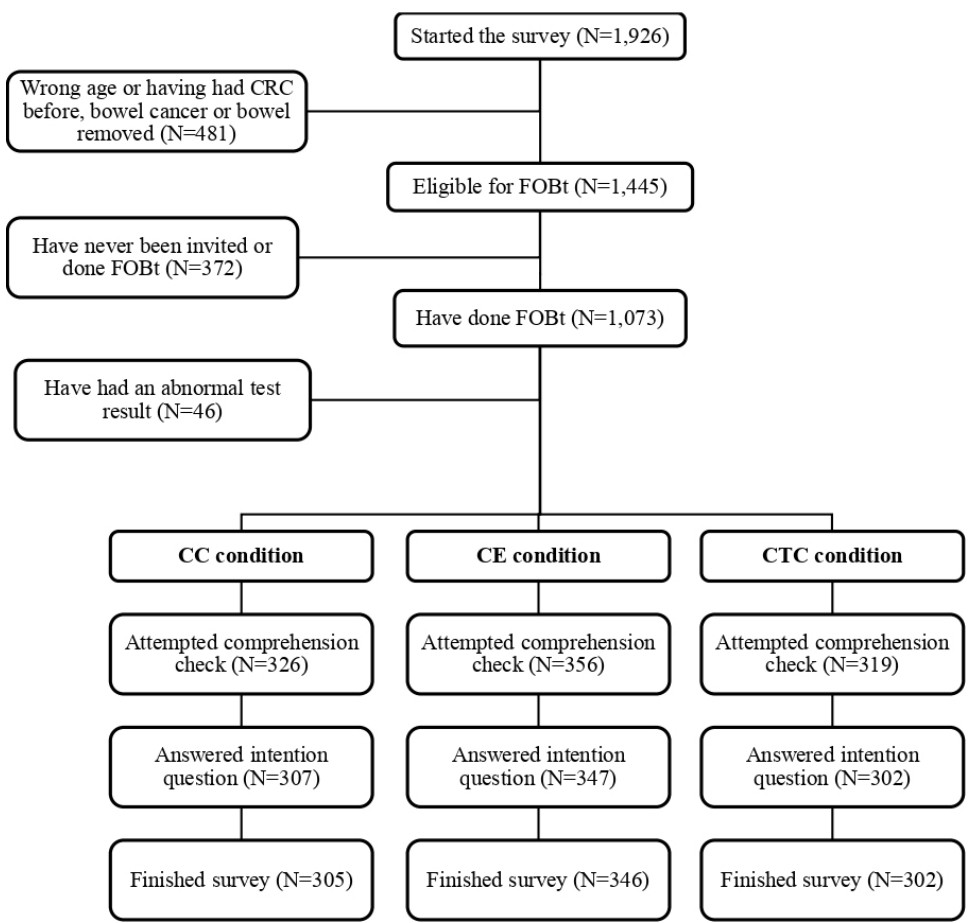

**Figure 1** Participant flow diagram. CC, colonoscopy; CE, capsule endoscopy; CRC, colorectal cancer; CTC, CT colonography; FOBt, faecal occult blood test.

### Intention to have the allocated test

A large majority (93.7%) intended to have their allocated test, although this was graded by ethnicity (61.5% vs 94.4%) and deprivation (2–3 markers: 87.9% vs 0 markers: 95.5%). The proportion of intenders was highest among those who were offered CTC, followed by those who were offered capsule, and finally those who were offered CC (96.7% vs 93.4% vs 91.8% respectively, $\chi^2=(2, n=953)=6.64, p=0.036$).

In the multivariable model, participants remained more likely to accept further investigation if the offered test was CTC compared with CC, after adjusting for covariates (table 2; OR 2.64, 95% CI 1.22 to 5.73, p=0.014). Offering CE did not significantly increase intentions compared with CC (OR 1.28, 95% CI 0.69 to 2.35, p=0.432). The multivariable analysis also revealed that intention to undergo follow-up tests was lower among BAME groups than White ethnic groups (OR 0.09, 95% CI 0.03 to 0.29, p<0.001), as well as more deprived individuals (1 marker: OR 0.45, 95% CI 0.25 to 0.84, p=0.012; 2–3 markers: OR 0.34, 95% CI 0.15 to 0.79, p=0.012).

### Preferences of disinclined study participants

In total, 58 (6.1%) study participants did not intend to have their offered follow-up test. Independently from the initial offer, most disinclined study participants would

either choose CE (n=23, 39.7%) or none of the tests (n=20, 34.5%). A Fisher's exact test demonstrated that preferences were not influenced by the initial random allocation (p=0.806).

### Perceived emotional barriers

Those who were randomised to CTC were less likely to agree that the test is off-putting (OR 0.66, 95% CI 0.47 to 0.94, p=0.022) or uncomfortable (OR 0.51, 95% CI 0.34 to 0.77, p=0.001) than those who were offered a CC. Participants who were offered CE as a follow-up test were less likely to perceive the test as uncomfortable (OR 0.11, 95% CI 0.07 to 0.16, p<0.001), embarrassing (OR 0.33, 95% CI 0. 23 to 0.48, p<0.001) or creating worries about the risks associated with the test (OR 0.70, 95% CI 0.51 to 0.97, p=0.031) than those who were offered a CC (table 3).

The table also shows that women were more likely to perceive the offered tests as off-putting (OR 1.41, 95% CI 1.06 to 1.88, p=0.018), embarrassing (OR 1.83, 95% CI 1.36 to 2.47, p<0.001) and causing worries about the risks (OR 1.46, 95% CI 1.11 to 1.91, p=0.007) and results of the test (OR 1.57, 95% CI 1.20 to 2.05, p=0.001).

Higher deprivation was associated with worrying about the risks associated with the test (OR 1.50, 95% CI 1.09 to 2.08, p=0.014) and numeracy was positively associated

**Table 1** Sociodemographic characteristics of the study population (n=953)

| | Colonoscopy condition (n=305) | | Capsule endoscopy condition (n=346) | | CT colonography condition (n=302) | | Overall (n=953) | |
|---|---|---|---|---|---|---|---|---|
| **Age** | | | | | | | | |
| 60–64 | 98 | 32.10% | 94 | 27.20% | 109 | 36.10% | 301 | 31.60% |
| 65–69 | 123 | 40.30% | 125 | 36.10% | 109 | 36.10% | 357 | 37.50% |
| 70–74 | 84 | 27.50% | 127 | 36.70% | 84 | 27.80% | 295 | 31.00% |
| **Gender** | | | | | | | | |
| Male | 151 | 49.50% | 201 | 58.10% | 167 | 55.30% | 519 | 54.50% |
| Female | 154 | 50.50% | 145 | 41.90% | 135 | 44.70% | 434 | 45.50% |
| **Ethnicity** | | | | | | | | |
| White | 301 | 98.70% | 341 | 98.60% | 298 | 98.70% | 940 | 98.60% |
| BAME | 4 | 1.30% | 5 | 1.40% | 4 | 1.30% | 13 | 1.40% |
| **Paid employment** | | | | | | | | |
| No | 221 | 72.50% | 260 | 75.10% | 235 | 77.80% | 716 | 75.10% |
| Yes | 84 | 27.50% | 86 | 24.90% | 67 | 22.20% | 237 | 24.90% |
| **Education** | | | | | | | | |
| No A levels | 119 | 39.00% | 125 | 36.10% | 112 | 37.40% | 356 | 37.40% |
| A levels or higher | 186 | 61.00% | 221 | 63.90% | 190 | 62.60% | 597 | 62.60% |
| **Car ownership** | | | | | | | | |
| No | 36 | 11.80% | 46 | 13.30% | 41 | 13.60% | 123 | 12.90% |
| Yes | 269 | 88.20% | 300 | 86.70% | 261 | 86.40% | 830 | 87.10% |
| **House ownership** | | | | | | | | |
| No | 44 | 14.40% | 46 | 13.30% | 30 | 9.90% | 120 | 12.60% |
| Yes | 261 | 85.60% | 300 | 86.70% | 272 | 90.10% | 833 | 87.40% |
| **Individual social deprivation*** | | | | | | | | |
| 0 markers | 223 | 73.10% | 242 | 69.90% | 221 | 73.20% | 686 | 72.00% |
| 1 marker | 58 | 19.00% | 80 | 23.10% | 63 | 20.80% | 201 | 21.10% |
| 2–3 markers | 24 | 7.90% | 24 | 6.90% | 18 | 5.90% | 66 | 6.90% |
| **Numeracy question** | | | | | | | | |
| Wrong | 163 | 53.40% | 180 | 52.00% | 151 | 50.00% | 494 | 51.80% |
| Correct | 142 | 46.60% | 166 | 48.00% | 151 | 50.00% | 459 | 48.20% |

*The three demographic questions on education, car and house ownership were used to calculate a proxy measure for socioeconomic deprivation. One point was given to an individual if their household did not have a car or van, if they had no formal qualifications and if they did not own their own home (1, 2). Scores, therefore, ranged from 0 to 3, with high scores indicating higher levels of social deprivation. Due to low frequencies, we collapsed the two groups that either had a score of 2 or three into one group called 2–3 markers.
BAME, Black, Asian and minority ethnic; CT, colonography.

with being afraid of the results (OR 1.32, 95% CI 1.01 to 1.72, p=0.39). Subgroup analysis did not reveal any significant interactions between these demographic variables and test allocation (online supplementary appendix 3).

### Perceived practical barriers

Study participants who were offered CE, instead of CC, were less likely to think that they would not have the time to do the test (OR 0.27, 95% CI 0.10 to 0.70, p=0.007). There were no other differences in practical barriers by test offered (see table 4). Health problems were cited as a barrier by participants with a higher number of deprivation markers (OR 2.54, 95% CI 1.27 to 5.07, p=0.009), those with a BAME background (OR 6.45, 95% CI 2.01 to 20.73, p=0.002) and by men (OR 1.85, 95% CI 2.94 to 1.16, p=0.009). Those from BAME groups (OR 9.37, 95% CI 2.23 to 39.36, p=0.002), and those who were in paid in employment (OR 2.42, 95% CI 1.09 to 5.38, p=0.03) were more likely to report not having enough time as a barrier to a follow-up investigation. Participants with a higher number of deprivation markers were also more likely to cite difficulties with transport (OR 3.52, 95% CI 1.96 to 6.31, p<0.001) as a barrier to having a follow-up investigation. Subgroup analysis did not reveal any significant interactions between these demographic

**Table 2** Association between test offered and wanting to have the test (n=953)

| | Unadjusted model | | Adjusted model | |
|---|---|---|---|---|
| | OR | 95% CI | OR | 95% CI |
| **Condition** | | | | |
| Colonoscopy | Ref. | | Ref. | |
| Capsule endoscopy | 1.25 | 0.70 to 2.26 | 1.28 | 0.69 to 2.35 |
| CT colonography | 2.61 | 1.23 to 5.53* | 2.64 | 1.22 to 5.73* |
| **Age** | | | | |
| 60–64 | | | Ref. | |
| 65–69 | | | 0.70 | 0.36 to 1.37 |
| 70–74 | | | 1.42 | 0.63 to 3.22 |
| **Gender** | | | | |
| Male | | | Ref. | |
| Female | | | 1.09 | 0.62 to 1.91 |
| Ethnicity | | | | |
| White | | | Ref. | |
| BAME | | | 0.09 | 0.03 to 0.29** |
| **Deprivation** | | | | |
| 0 makers | | | Ref. | |
| 1 marker | | | 0.45 | 0.25 to 0.84* |
| 2–3 markers | | | 0.34 | 0.15 to 0.79* |
| **Paid employment** | | | | |
| No | | | Ref. | |
| Yes | | | 0.87 | 0.45 to 1.67 |
| Numeracy task | | | | |
| Wrong | | | Ref. | |
| Correct | | | 1.58 | 0.89 to 2.78 |

Logistic regression models presented.
*P<0.05; **P<0.01.
BAME, Black, Asian and minority ethnic; CT, colonography.

variables and test allocation (online supplementary appendix 3).

## DISCUSSION

This study investigated whether offering alternative follow-up tests increases intention to undergo further investigation in the context of CRC screening. The results of our study show that, while the vast majority of individuals would accept any test, offering CTC instead of CC may increase acceptance of follow-up tests. Findings from our study suggest that this is due to individuals perceiving CTC as less 'off-putting' and 'uncomfortable' than CC. No differences were found in intention between those randomised to CE compared with CC despite CE appearing less uncomfortable, less embarrassing, causing less worry about the risks associated with the test and taking test time. However, for the small number of participants who did not intend to take the offered test, CE appeared to be the preferred test when presented with all options.

The findings from this study are supported by two trials: one in in France, by Pioche et al and one in Italy, by Sali et al.[32 33] Both studies randomised patients with abnormal FOBt results who had refused the offer of a CC to be invited for another test. Pioche et al found that CTC was more appealing than CE (7.4% vs 5.0% having the investigation) with no differences found in clinical outcomes. The authors concluded that those who refuse CC are difficult to recruit into the screening programme and that simply offering a different test is not effective. Sali et al also found that invitation to CTC was more preferable, but in comparison to CC (35.7% vs 14.1% attending). However, these trials were carried out in France and Italy and may not be generalisable to a UK population and the National Health Service (NHS) screening programme. Importantly, as these studies focused on patients who had already refused CC it looked at preferences in a preselected sample. This is reflected in the small sample sizes in both studies with only 50–100 participants in each comparison group, making it difficult to draw conclusions about the population-level benefits of offering alternative tests.

Our study observed a lower likelihood of ethnic minorities to accept any follow-up test. This is supported by a recent systematic review which found that non-white participants were less likely to attend in 9 out of 10 studies investigating incomplete diagnostic follow-up after a positive bowel screening test.[11] Analysis of barriers suggest that other health problems and not having enough time may be additional contributing factors to intention for BAME participants. Previous research has identified embarrassment as a potential barrier to CRC screening, although this was not found in this study.[34] We also found that not having enough time, and difficulties arranging transport are barriers to screening for people in employment, those with a higher level of deprivation and those from a BAME background. This is consistent with evidence from an American study, which showed that the location of the CC appointment and the availability of evening and weekend appointments are important factors limiting uptake.[35] These findings should be interpreted with care due to the very small number of BAME participants in the entire sample and the heterogeneity of the group. Future research should aim to recruit a larger and more representative BAME sample so findings can be presented for each ethnic group.

While our study provides support that individuals perceive follow-up tests differently, we only offered one test at random. Future research should present all three tests to participants in a random order to analyse whether offering individuals the choice between different tests increases intentions.[22] This could also include investigation of other diagnostic tests such as stool DNA tests.[36]

**Table 3** Association between test offered and emotional barrier items (n=953)

| | Off-putting | | Uncomfortable | | Embarrassing | | Worry about risks of test | | Afraid of results | | Worry about cancer | |
|---|---|---|---|---|---|---|---|---|---|---|---|---|
| | OR | 95% CI | OR | 95% CI | OR | 95% CI | OR | 95% CI | OR | 95% CI | OR | 95% CI |
| Condition | | | | | | | | | | | | |
| Colonoscopy | Ref. | | Ref. | | Ref. | | Ref. | | Ref. | | Ref. | |
| Capsule endoscopy | 0.75 | 0.53 to 1.047 | 0.11 | 0.07 to 0.16** | 0.33 | 0.23 to 0.48** | 0.70 | 0.51 to 0.97* | 0.75 | 0.54 to 1.03 | 0.82 | 0.59 to 1.13 |
| CT colonography | 0.66 | 0.47 to 0.94* | 0.51 | 0.34 to 0.77** | 0.72 | 0.51 to 1.02 | 0.73 | 0.52 to 1.02 | 0.92 | 0.66 to 1.28 | 1.15 | 0.83 to 1.60 |
| Age | | | | | | | | | | | | |
| 60–64 | Ref. | | Ref. | | Ref. | | Ref. | | Ref. | | Ref. | |
| 65–69 | 0.99 | 0.69 to 1.41 | 0.91 | 0.62 to 1.32 | 0.81 | 0.56 to 1.16 | 0.85 | 0.61 to 1.19 | 0.68 | 0.48 to 0.94* | 0.67 | 0.48 to 0.93* |
| 70–74 | 1.05 | 0.72 to 1.54 | 0.89 | 0.60 to 1.33 | 0.88 | 0.59 to 1.31 | 0.88 | 0.62 to 1.27 | 0.71 | 0.49 to 1.01 | 0.74 | 0.52 to 1.06 |
| Gender | | | | | | | | | | | | |
| Male | Ref. | | Ref. | | Ref. | | Ref. | | Ref. | | Ref. | |
| Female | 1.41 | 1.06 to 1.88* | 1.03 | 0.76 to 1.39 | 1.83 | 1.36 to 2.47** | 1.46 | 1.11 to 1.91** | 1.57 | 1.20 to 2.05** | 1.05 | 0.801 to 1.38 |
| Ethnicity | | | | | | | | | | | | |
| White | Ref. | | Ref. | | Ref. | | Ref. | | Ref. | | Ref. | |
| BAME | 2.22 | 0.73 to 6.75 | 1.22 | 0.33 to 4.55 | 1.19 | 0.34 to 4.19 | 2.02 | 0.66 to 6.17 | 3.97 | 0.86 to 18.33 | 1.76 | 0.58 to 5.34 |
| Deprivation | | | | | | | | | | | | |
| 0 markers | Ref. | | Ref. | | Ref. | | Ref. | | Ref. | | Ref. | |
| 1 marker | 1.06 | 0.75 to 1.50 | 1.36 | 0.93 to 1.97 | 0.99 | 0.69 to 1.43 | 1.50 | 1.09 to 2.08* | 1.36 | 0.97 to 1.89 | 1.34 | 0.97 to 1.86 |
| 2–3 markers | 1.43 | 0.84 to 2.44 | 0.65 | 0.36 to 1.15 | 1.29 | 0.73 to 2.26 | 1.26 | 0.75 to 2.14 | 0.68 | 0.41 to 1.14 | 0.99 | 0.58 to 1.680 |
| Paid employment | | | | | | | | | | | | |
| No | Ref. | | Ref. | | Ref. | | Ref. | | Ref. | | Ref. | |
| Yes | 0.99 | 0.70 to 1.40 | 1.45 | 1.00 to 2.11 | 1.41 | 1.00 to 2.01 | 1.11 | 0.80 to 1.53 | 1.19 | 0.86 to 1.65 | 1.01 | 0.73 to 1.40 |
| Numeracy task | | | | | | | | | | | | |
| Wrong | Ref. | | Ref. | | Ref. | | Ref. | | Ref. | | Ref. | |
| Correct | 1.03 | 0.79 to 1.37 | 0.88 | 0.66 to 1.19 | 1.13 | 0.84 to 1.52 | 1.08 | 0.83 to 1.42 | 1.32 | 1.01 to 1.72* | 1.22 | 0.94 to 1.59 |

Logistic regression models presented

*P<0.05; **P<0.01.

BAME, Black, Asian and minority ethnic; CT, colonography.

**Table 4** Association between study condition and practical barrier items (n=953)

| | No time | | Other problems | | Difficulties with transport | | Health problems | |
|---|---|---|---|---|---|---|---|---|
| | OR | 95% CI | OR | 95% CI | OR | 95% CI | OR | 95% CI |
| Condition | | | | | | | | |
| Colonoscopy | Ref. | | Ref. | | Ref. | | Ref. | |
| Capsule endoscopy | 0.27 | 0.10–0.70** | 0.75 | 0.52 to 1.09 | 0.76 | 0.48 to 1.18 | 0.70 | 0.42 to 1.17 |
| CT colonography | 0.46 | 0.19 to 1.09 | 0.80 | 0.55 to 1.18 | 0.64 | 0.40 to 1.04 | 0.58 | 0.33 to 1.01 |
| Age | | | | | | | | |
| 60–64 | Ref. | | Ref. | | Ref. | | Ref. | |
| 65–69 | 1.06 | 0.43 to 2.63 | 0.91 | 0.62 to 1.35 | 0.77 | 0.49 to 1.23 | 0.92 | 0.53 to 1.58 |
| 70–74 | 1.28 | 0.48 to 3.39 | 1.03 | 0.68 to 1.55 | 0.65 | 0.39 to 1.10 | 0.80 | 0.44 to 1.46 |
| Gender | | | | | | | | |
| Male | Ref. | | Ref. | | Ref. | | Ref. | |
| Female | 0.73 | 0.35 to 1.55 | 1.18 | 0.86 to 1.61 | 1.33 | 0.91 to 1.96 | 0.54 | 0.34 to 0.86** |
| Ethnicity | | | | | | | | |
| White | Ref. | | Ref. | | Ref. | | Ref. | |
| BAME | 9.37 | 2.23 to 39.36** | 2.28 | 0.73 to 7.10 | 3.32 | 0.99 to 11.13 | 6.45 | 2.01 to 20.73** |
| Deprivation | | | | | | | | |
| 0 markers | Ref. | | Ref. | | Ref. | | Ref. | |
| 1 marker | 1.40 | 0.59 to 3.32 | 1.11 | 0.76 to 1.61 | 1.65 | 1.06 to 2.58* | 1.35 | 0.79 to 2.29 |
| 2–3 markers | 2.41 | 0.78 to 7.49 | 1.37 | 0.77 to 2.44 | 3.52 | 1.96 to 6.31** | 2.54 | 1.27 to 5.07** |
| Paid employment | | | | | | | | |
| No | Ref. | | Ref. | | Ref. | | Ref. | |
| Yes | 2.42 | 1.09 to 5.38* | 0.97 | 0.66 to 1.42 | 0.88 | 0.55 to 1.41 | 0.95 | 0.56 to 1.63 |
| Numeracy task | | | | | | | | |
| Wrong | Ref. | | Ref. | | Ref. | | Ref. | |
| Correct | 0.60 | 0.28 to 1.26 | 0.83 | 0.61 to 1.14 | 0.74 | 0.50 to 1.09 | 0.96 | 0.61 to 1.49 |

Logistic regression models presented.
*P<0.05; **P<0.01.
BAME, Black, Asian and minority ethnic; CT, colonography.

## Strengths and limitations

Our study has several limitations. First, our study used hypothetical scenarios and non-representative online study samples. Although the vignettes were developed in collaboration with clinicians, it was not possible in the space available to fully explain the level of burden associated with each test which may have biased responses. For example, in the CE vignette we stated that after the procedure it was possible to return to normal daily activities, when in reality the patient may be required to take booster laxatives and will have to return another day to have the recording analysed. It is possible that CE emerged as the second most preferred test due to the relatively positive description of the procedure.

Furthermore, the experiments lacked behavioural validation, in that they only measured intentions and choice in a hypothetical setting. The next step would be to test the effect of offering alternative follow-up tests under more ecologically valid conditions in a randomised controlled trial.

A strength of our study is that we used comprehension checks to ensure that all study participants sufficiently engaged with the provided information about the follow-up test.

## CONCLUSIONS

While the results from this randomised online experiment show a high acceptance of the current practice of the NHS Bowel Screening Programme to invite individuals who have an abnormal FOBt result for a follow-up CC, our study also provides some evidence that offering CTC, instead of CC, may increase the total number of participants undertaking a follow-up test. Although CTC and CE are not currently endorsed as a screening modalites, our study provides some evidence that for high-risk and hard-to-engage groups, such as those who have a positive FOBt but refuse follow-up, offering alternative tests may increase intentions to engage with medical professionals and receive appropriate care.

**Acknowledgements** We would like to acknowledge the contribution to the study materials by Professor Owen Epstein, Tim Rayne and Dr. Andrew Plumb.

**Contributors** AK, STS and CvW developed the study concept. All authors contributed to the study design. AK, STS and CvW performed the data analysis and interpretation. AK and STS drafted the manuscript, RK and CvW provided critical revisions. All authors approved the final version of the manuscript for submission.

**Funding** This report presents independent research commissioned and funded by the National Institute for Health Research (NIHR) Policy Research Programme, conducted through the Policy Research Unit in Cancer Awareness, Screening and Early Diagnosis, PR-PRU-1217–21601. RK is supported by a Cancer Research UK Population Research Fellowship (C68512/A28209).

**Disclaimer** The views expressed are those of the authors and not necessarily those of the NIHR, the Department of Health and Social Care or its arm's length bodies, or other Government Departments.

**Competing interests** None declared.

**Patient consent for publication** Obtained.

**Ethics approval** This research project was approved by UCL Research ethics committee (approval number 14687/001). All participants were asked to give explicit written consent for their data to be used and published as part of this research project before taking part in this study.

**Provenance and peer review** Not commissioned; externally peer reviewed.

**Data availability statement** All data relevant to the study are included in the article or uploaded as online supplementary information. All data files and materials are publicly available via the Open Science Framework and can be accessed at https://accounts.osf.io/login?service=https://osf.io/fx69t/?view_only=46ba 2185c26940a89c19bde28da40a9c.&view_only=46ba2185c26940a89c19bde2 8da40a9c

**ORCID iDs**
Aradhna Kaushal http://orcid.org/0000-0002-3815-0624
Robert Kerrison http://orcid.org/0000-0002-8900-749X
Christian von Wagner http://orcid.org/0000-0002-7971-0691

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
