## [Reviewer comments · BMJ Open]

ARTICLE DETAILS

TITLE (PROVISIONAL)	Preferences for different diagnostic modalities to follow up abnormal colorectal cancer screening results: a hypothetical vignette study
AUTHORS	Kaushal, Aradhna; Stoffel, Sandro; Kerrison, Robert; von Wagner, Christian

VERSION 1 - REVIEW

REVIEWER	Carlo Senore Epidemiology and screening Unit - CPO University Hospital Città della Salute e della Scienza Turin, Italy
REVIEW RETURNED	22-Nov-2019

GENERAL COMMENTS	The authors are reporting the results of a study aimed at comparing the potential impact of offering alternative assessment tests to screenees with a positive FOBT result. Eligible subjects having attended previous FOBT invitations were asked to indicate in the context of an online experiment whether their intention to attend the assessment examination when offered CTC or CE. The research question addressed in this study is highly relevant as the proportion of FOBT positive subjects who do not comply with the recommendation to undergo a colonoscopy assessment is high (often between 30 and 20%). Considering the high PPV for advanced neoplasia of FOBT tests, this can result in a substantial reduction of the potential protective effect of screening. The study was well designed and the description of the results is clear. The main limitations are mentioned in the discussion. The choice to contact people who had already attended screening with FOBT in the national program represent a strength of the study, although it would be relevant to know how many subjects were invited to participate in the study. Even if demographic data about the entire target sample might be limited, the information might be useful to assess the potential for self selection of responders. The analyses of the perceived emotional and practical barriers, as well as the reasons of non-attendance are providing useful information. The authors should however consider that, in addition to the study they mention in the discussion considering the offer of CE, there are already studies offering, for example, CT colonography to subjects refusing TC following a positive FIT, in a real world situation. Also, reasons for non-attendance to, and barriers limiting compliance with, TC assessment have been explored in other surveys. They should justify the additional value of their study, as compared to information already available from these studies.
---

	The authors should provide some justification for their choice to include also a test (CE) which is still under evaluation and not routinely recommended in the screening guidelines, or even to complete a TC. Also, the description of the test procedure used in the vignette would seem over-optimistic. The description of the bowel preparation protocol “ ... you need to restrict your diet and take a strong laxative” does not fully convey the actual burden of the protocols adopted in CE studies. Similarly, saying that the person can continue his/her daily activities after having ingested the capsule does not really correspond to the actual practice with CE, requiring monitoring of capsule progression and ingestion of 1-2 laxative boosters. This way to present the test might explain while it was considered a better option than TC and a choice among those refusing all tests.
--	---

REVIEWER	William E Barlow Cancer Research & Biostatistics United States
REVIEW RETURNED	14-Feb-2020

GENERAL COMMENTS	This manuscript describes hypothetical completion of colorectal cancer testing after a positive fit based on randomized assignment to three vignettes corresponding to different diagnostic procedures. There was some preference for CT colonography (CTC) compared to more invasive procedures such as colonoscopy (CC) and capsule endoscopy (CE). The manuscript is very well written and the analytic work is clear so the comments will be limited to more minor issues. Overall the manuscript is quite interesting and a contribution to the literature.  1. The provided scenario materials are even-handed and describe similar risks of missing cancers by the particular test. However, in the U.S. the Preventive Services Task Force does not endorse CTC as a screening modality. In this particular study it is being used as the diagnostic test following a positive FOBT, not initial screening, but the concerns of radiation exposure and shorter protection interval (5 years vs. 10 years for CC) make it less desirable even if more convenient. Furthermore, a positive CTC just leads to another CC so the efficiency is less. 2. We found a 5.4% positivity rate for FOBT/FIT in those aged 65-75 so the 2% positivity rate in the BCSP is impressive. We also found 26% lacked timely follow-up of positives in that age range so again the BCSP is superior in follow-up. Reference: Barlow WE, Beaber EF, Geller BM, et al. Evaluating screening participation, follow-up and outcomes for breast, cervical and colorectal cancer in the PROSPR consortium. J Natl Cancer Inst. 2019 Jul 11. PubMed PMID: 31292633. {we are not angling to include this reference in the manuscript...} 3. The vignettes look very clear and there is a nice check on understanding embedded in the survey. Participation rates on the survey are high. Since they were conditional on having completed a previous FOBT that was negative, there may be some bias toward greater participation than in a general population being offered screening for the first time. 4. It would have been a much lengthier survey, but presenting all three vignettes in random order would have yielded strong results about whether test type really influences participation. 5. The sample size calculation was unclear. Was it powered to find a
---

	5% difference anywhere among all three groups. The test being used (2df) is testing any difference among the three groups. 6. Since randomization is used to assign participants, hypothesis testing of differences by factors (Table 1) as statistical significance is irrelevant. The null hypothesis must be true due to randomization and only chance differences will be observed. Consider deletion of the p-values in Table 1. 7. Participants who are non-British white are lumped together as BAME, but there are only 13 individuals in this group. This number is too small to be modeled and the group is too heterogenous to be interpreted. Consider removal of that variable from all analyses (keeping the 13 individuals). It is statistically significant in places, but not interpretable. 8. Table 2. It is interesting that the testing method has the same effect whether adjusted or not (probably due to randomization). OR's can have only two digits to the right of the decimal place as in the text. 9. "Intention to have the allocated test" section – It alludes to results in Table 1, but outcomes are not shown in Table 1. Can the raw percentages also be displayed in Tables 2, 3, and 4 but only for the conditions? 10. Tables may need more extensive captions.
--	---

VERSION 1 – AUTHOR RESPONSE

Reviewer: Carlo Senore

The authors are reporting the results of a study aimed at comparing the potential impact of offering alternative assessment tests to screenees with a positive FOBT result. Eligible subjects having attended previous FOBT invitations were asked to indicate in the context of an online experiment whether their intention to attend the assessment examination when offered CTC or CE.

The research question addressed in this study is highly relevant as the proportion of FOBT positive subjects who do not comply with the recommendation to undergo a colonoscopy assessment is high (often between 30 and 20%). Considering the high PPV for advanced neoplasia of FOBT tests, this can result in a substantial reduction of the potential protective effect of screening.

1. The study was well designed and the description of the results is clear. The main limitations are mentioned in the discussion. The choice to contact people who had already attended screening with FOBT in the national program represent a strength of the study, although it would be relevant to know how many subjects were invited to participate in the study. Even if demographic data about the entire target sample might be limited, the information might be useful to assess the potential for self selection of responders.

We do not know how many participants are invited to the study. Recruitment was carried out by Dynata who invite eligible participants from their panel.

2. The analyses of the perceived emotional and practical barriers, as well as the reasons of non-attendance are providing useful information. The authors should however consider that, in addition to the study they mention in the discussion considering the offer of CE, there are already studies offering, for example, CT colonography to subjects refusing TC following a positive FIT, in a real world situation. Also, reasons for non-attendance to, and barriers limiting compliance with, TC assessment have been explored in other surveys. They should justify the additional value of their study, as compared to information already available from these studies.

We have revised our discussion to show the additional value of this study compared to existing research.

“This trial was carried out in France and may not be generalisable to a UK population and the NHS screening programme. Importantly, as the study focused on patients who had already refused CC it looked at preferences in a pre-selected sample. This is reflected in the small sample size with less than 50 participants in each comparison group which makes it difficult to say for certain that offering alternative tests would not be of any benefit at a population level.

”

3. The authors should provide some justification for their choice to include also a test (CE) which is still under evaluation and not routinely recommended in the screening guidelines, or even to complete a TC.

This was an exploratory piece of research and while we are aware that CE is not recommended as a screening tool, they may be of some benefit to the small minority of people who have a positive FOBT/FIT result but for reasons otherwise unknown refuse a colonoscopy. We have revised our introduction section to clarify this point.

“CTC and CE could be offered individuals who decline to have a CC after a positive FOBt ...Although CTC and CE are not currently endorsed as a screening methods, it is possible that offering alternative tests for those who do stop engaging with the screening programme may increase intentions to have further investigations.”

4. Also, the description of the test procedure used in the vignette would seem over-optimistic. The description of the bowel preparation protocol “ ... you need to restrict your diet and take a strong laxative” does not fully convey the actual burden of the protocols adopted in CE studies. Similarly, saying that the person can continue his/her daily activities after having ingested the capsule does not really correspond to the actual practice with CE, requiring monitoring of capsule progression and ingestion of 1-2 laxative boosters. This way to present the test might explain while it was considered a better option than TC and a choice among those refusing all tests.

It was challenging to provide accurate and succinct information on each test in the form of an online survey. We agree that although complete burden of each test is not conveyed, the vignettes do communicate the relative advantages and risks of each test and were checked by clinicians who are experts in administering these tests. We have now acknowledged this limitation in the discussion section.

“Although the vignettes were developed in collaboration with clinicians, it was not possible in the space available to fully explain the level of burden associated with each test which may have biased responses.”

Reviewer: William E Barlow

This manuscript describes hypothetical completion of colorectal cancer testing after a positive FIT based on randomized assignment to three vignettes corresponding to different diagnostic procedures. There was some preference for CT colonography (CTC) compared to more invasive procedures such as colonoscopy (CC) and capsule endoscopy (CE). The manuscript is very well written and the analytic work is clear so the comments will be limited to more minor issues. Overall the manuscript is quite interesting and a contribution to the literature.

1. The provided scenario materials are even-handed and describe similar risks of missing cancers by the particular test. However, in the U.S. the Preventive Services Task Force does not endorse CTC as a screening modality. In this particular study it is being used as the diagnostic test following a

positive FOBT, not initial screening, but the concerns of radiation exposure and shorter protection interval (5 years vs. 10 years for CC) make it less desirable even if more convenient. Furthermore, a positive CTC just leads to another CC so the efficiency is less.

“CTC and CE could be offered individuals who decline to have a CC after a positive FOBT ...Although CTC and CE are not currently endorsed as a screening methods, it is possible that offering alternative tests for those who do stop engaging with the screening programme may increase intentions to have further investigations.”

“Although CTC and CE are not currently endorsed as a screening modalities, our study provides some evidence that for high-risk and hard-to-engage groups, such as those who have a positive FOBT but refuse follow-up, offering alternative tests may increase intentions to have engage with medical professionals and receive appropriate care.”

2. We found a 5.4% positivity rate for FOBT/FIT in those aged 65-75 so the 2% positivity rate in the BCSP is impressive. We also found 26% lacked timely follow-up of positives in that age range so again the BCSP is superior in follow-up.

Reference: Barlow WE, Beaber EF, Geller BM, et al. Evaluating screening participation, follow-up and outcomes for breast, cervical and colorectal cancer in the PROSPR consortium. J Natl Cancer Inst. 2019 Jul 11. PubMed PMID: 31292633.

{we are not angling to include this reference in the manuscript...}

Thank you for your comment. As positivity is contingent on decisions within the programmes such as thresholds we do not feel comfortable to comment on this directly in our manuscript. We agree that participation in diagnostic follow up compares well with other programmes, however it still poses an important issue which we hope to begin to address with this study.

3. The vignettes look very clear and there is a nice check on understanding embedded in the survey. Participation rates on the survey are high. Since they were conditional on having completed a previous FOBT that was negative, there may be some bias toward greater participation than in a general population being offered screening for the first time.

It is possible that the sample is biased towards those who are more likely to take part in screening, however our population of interest are those who have taken an FOBT, had a positive result but have not had any follow-up.

4. It would have been a much lengthier survey, but presenting all three vignettes in random order would have yielded strong results about whether test type really influences participation.

We agree it would have been ideal to present all three vignettes in random order but this may have been at the expense of a high completion rate. In our experience, there is a risk for participants to drop-out of online surveys if there is a large amount of reading and/or comprehension checks. We have suggested a comparison of all three tests for future research.

“Future research should present all three tests to participants in a random order to analyse whether offering individuals the choice between different tests increases intentions.”

5. The sample size calculation was unclear. Was it powered to find a 5% difference anywhere among all three groups. The test being used (2df) is testing any difference among the three groups.

Yes, the sample was powered to find a difference of 5% for each pair-wise comparison. We have revised the text to clarify this.

“We calculated that we needed approximately 300 participants per condition to detect differences of at least 5 percentage points in proportion of intenders effect size between any of the three conditions, with a power of 80% and an alpha value of 0.05”

6. Since randomization is used to assign participants, hypothesis testing of differences by factors (Table 1) as statistical significance is irrelevant. The null hypothesis must be true due to randomization and only chance differences will be observed. Consider deletion of the p-values in Table 1.

We have removed the p-values from Table.1

7. Participants who are non-British white are lumped together as BAME, but there are only 13 individuals in this group. This number is too small to be modeled and the group is too heterogenous to be interpreted. Consider removal of that variable from all analyses (keeping the 13 individuals). It is statistically significant in places, but not interpretable.

We agree that the results of the BAME group are not interpretable due to small number and the heterogeneity of this group. However, we feel that this finding is supported by previous research and warrants further research. We have revised our discussion section to clarify this.

“These findings should be interpreted with care due to the very small number of BAME participants in the entire sample and the heterogeneity of the group. Future research should aim to recruit a larger and more representative BAME sample so findings can be presented for each ethnic group.”

8. Table 2. It is interesting that the testing method has the same effect whether adjusted or not (probably due to randomization). OR's can have only two digits to the right of the decimal place as in the text.

All the figures in Table 2. Are now to 2 d.f.

9. “Intention to have the allocated test” section – It alludes to results in Table 1, but outcomes are not shown in Table 1. Can the raw percentages also be displayed in Tables 2, 3, and 4 but only for the conditions?

This was an error and has been corrected.

Adding the raw percentages in Tables 2, 3 and 4 is not possible due to space constraints. These figures are available in the supplementary materials and we have included these in the main text for headline findings.

10. Tables may need more extensive captions.

Table captions have been revised.

VERSION 2 – REVIEW

REVIEWER	Carlo Senore Epidemiology and screening unit - CPO University hospital Città della Salute e della Scienza Turin, Italy
REVIEW RETURNED	07-Apr-2020

GENERAL COMMENTS	The authors provided adequate responses to most of the points
---

	raised by the reviewers, However, two issues still need to be considered. Clearly, vignettes cannot be too long and some messages need to be simplified, but, also considering that the results might have been different when presenting the characteristics of the CE slightly different (more realistic description), I still think that this should be mentioned more explicitly in the discussion, when commenting the results. An over-optimistic description of CE method (it could be Ok to shorten the description of the bowel prep, but it is unrealistic, at the moment, to tell that the person can continue his/her daily activities after having ingested the capsule). As this emerged as an acceptable method, it should be acknowledged that the framing of the message might have influenced the results. I still think that references to real-world experiences, which, incidentally, confirm that offering an alternative test to CT refusers can increase compliance, should be added. There is at least a study in performed in the context of the Florence program in Italy, showing an increase in the proportion of FIT positive subjects having a complete examination of the colon when offering CTC to TC refusers.
--	--

REVIEWER	William E Barlow, PhD Cancer Research & Biostatistics Seattle WA United States
REVIEW RETURNED	27-Mar-2020

GENERAL COMMENTS	I have no further comments on the revision and believe my points have been adequately addressed.
--

VERSION 2 – AUTHOR RESPONSE

Reviewer: Carlo Senore

The authors provided adequate responses to most of the points raised by the reviewers, however, two issues still need to be considered.

1. Clearly, vignettes cannot be too long and some messages need to be simplified, but, also considering that the results might have been different when presenting the characteristics of the CE slightly different (more realistic description), I still think that this should be mentioned more explicitly in the discussion, when commenting the results. An over-optimistic description of CE method (it could be Ok to shorten the description of the bowel prep, but it is unrealistic, at the moment, to tell that the person can continue his/her daily activities after having ingested the capsule). As this emerged as an acceptable method, it should be acknowledged that the framing of the message might have influenced the results.

Thank you for your suggestion. We have revised the discussion section to more fully acknowledge the limitation caused by the optimistic description of CE.

"Although the vignettes were developed in collaboration with clinicians, it was not possible in the space available to fully explain the level of burden associated with each test which may have biased responses. For example, in the CE vignette we stated that after the procedure it was possible to return to normal daily activities, when in reality the patient may be required to take booster laxatives and will have to return another day to have the recording analysed. It is possible that CE emerged as the second most preferred test due to the relatively positive description of the procedure."

2. I still think that references to real-world experiences, which, incidentally, confirm that offering an alternative test to CT refusers can increase compliance, should be added. There is at least a study in performed in the context of the Florence program in Italy, showing an increase in the proportion of FIT positive subjects having a complete examination of the colon when offering CTC to TC refusers.

Thank you for directing us to this relevant and interesting paper. We have added this paper by Sali et al (2013) to our comparison with other literature.

"The findings from this study are supported by two trials: one in France, by Pioche et al. (2018) and one in Italy, by Sali et al. (2013). 32, 33 Both studies randomised patients with abnormal FOBt results who had refused the offer of a colonoscopy to be invited for another test. Pioche and colleagues found that CTC was more appealing than CE (7.4% vs. 5.0% having the investigation) with no differences found in clinical outcomes. The authors concluded that those who refuse CC are difficult to recruit into the screening programme and that simply offering a different test is not effective. Sali and colleagues also found that invitation to CTC was more preferable, but in comparison to CC (35.7% vs. 14.1% attending). These trials were carried out in France and Italy and may not be generalisable to a UK population and the NHS screening programme. Importantly, as the studies focused on patients who had already refused CC it looked at preferences in a pre-selected sample. This is reflected in the small sample sizes in both studies with only 50-100 participants in each comparison group, making it difficult to draw conclusions about the population-level benefits of offering alternative tests."

VERSION 3 - REVIEW

REVIEWER	Carlo Senore Epidemiology and screening unit - CPO. University hospital Città della Salute e della Scienza, Turin. Italy
REVIEW RETURNED	05-May-2020

GENERAL COMMENTS	I have no further comment The authors have carefully addressed the issues raised by the reviewers The limitations are now adequately discussed
--